# The Deubiquitinating Enzyme USP20 Regulates the TNFα-Induced NF-κB Signaling Pathway through Stabilization of p62

**DOI:** 10.3390/ijms21093116

**Published:** 2020-04-28

**Authors:** Jihoon Ha, Minbeom Kim, Dongyeob Seo, Jin Seok Park, Jaewon Lee, Jinjoo Lee, Seok Hee Park

**Affiliations:** Department of Biological Sciences, Sungkyunkwan University, Suwon 16419, Korea; wlgns293@skku.edu (J.H.); minbum@skku.edu (M.K.); sdy2089@skku.edu (D.S.); ezs7777@naver.com (J.S.P.); 89jaelee@skku.edu (J.L.); ljj0441@gmail.com (J.L.)

**Keywords:** ubiquitin-specific protease 20, p62, tumor necrosis factor α, nuclear factor-κB, cell survival, apoptosis

## Abstract

p62/sequestosome-1 is a scaffolding protein involved in diverse cellular processes such as autophagy, oxidative stress, cell survival and death. It has been identified to interact with atypical protein kinase Cs (aPKCs), linking these kinases to NF-κB activation by tumor necrosis factor α (TNFα). The diverse functions of p62 are regulated through post-translational modifications of several domains within p62. Among the enzymes that mediate these post-translational modifications, little is known about the deubiquitinating enzymes (DUBs) that remove ubiquitin chains from p62, compared to the E3 ligases involved in p62 ubiquitination. In this study, we first demonstrate a role of ubiquitin-specific protease USP20 in regulating p62 stability in TNFα-mediated NF-κB activation. USP20 specifically binds to p62 and acts as a positive regulator for NF-κB activation by TNFα through deubiquitinating lysine 48 (K48)-linked polyubiquitination, eventually contributing to cell survival. Furthermore, depletion of USP20 disrupts formation of the atypical PKCζ-RIPK1-p62 complex required for TNFα-mediated NF-κB activation and significantly increases the apoptosis induced by TNFα plus cycloheximide or TNFα plus TAK1 inhibitor. These findings strongly suggest that the USP20-p62 axis plays an essential role in NF-κB-mediated cell survival induced by the TNFα-atypical PKCζ signaling pathway.

## 1. Introduction

Ever since p62—also known as sequestosome-1 (SQSTM1)—was identified as a scaffold protein for atypical protein kinase C (aPKC) [1,2], it has been recognized as a signaling hub for diverse cellular signaling pathways involved in autophagy, oxidative stress, cell survival and cell death [3,4]. These diverse functions seem to be due to scaffolding activities mediated by multiple domains within p62, which interact with various binding partners [4]. Based on the multiple domains of p62, much attention has been paid to the autophagy process. In particular, p62 is primarily degraded through selective autophagy [5,6], although it can also be degraded by the proteasome or endosomal-related autophagy [7,8,9], and thus p62 levels are commonly measured to monitor autophagic flux [10,11]. p62 is also involved in other cellular processes such as the NF-κB signaling pathway through aPKCs [1,2], receptor interacting protein kinase-1 (RIPK1) [12] and tumor necrosis factor receptor-associated factor 6 (TRAF6) [13], adipogenesis through interaction with extracellular signal-regulated kinase 1 (ERK1) [14], the antioxidant response through interaction with Keap1 [15], apoptosis through caspase-8 [16] and nutrient sensing through interactions with Raptor in the mammalian target of rapamycin (mTOR) pathway [17].

Cell context-dependent roles of p62 in these diverse processes require post-translational modifications of the p62 protein such as phosphorylation and ubiquitination by specific binding partners [18]. Extensive studies demonstrate p62 phosphorylation by many kinases such as protein kinase A (PKA) [19], cyclin-dependent kinase 1 (CDK1) [20], Unc51-like kinase 1 (ULK1) [21] and AMP-activated protein kinase (AMPK) [22], but relatively less has been reported on the ubiquitination or deubiquitination of p62. Recently, the Keap1-Cullin3 complex was reported to ubiquitinate lysine 420 in the UBA domain of p62 and this ubiquitination was found to promote p62 sequestering activity and autophagic degradation [23]. In contrast, the E3 ubiquitin ligase TRIM21 directly interacts with and ubiquitinates p62 at lysine 7 within the PB1 domain, subsequently inhibiting its sequestration function [24]. In addition, the E3 ubiquitin ligase RNF166 was reported to facilitate p62 recruitment during antibacterial autophagy through polyubiquitination of lysine 91 and 189 of p62 [25]. Non-degradative ubiquitination of p62 by the E3 ligase RNF26 is required for the recruitment of endocytic vesicles to the perinuclear region [26]. Compared to the processes ubiquitinating p62, little is known about the deubiquitinating enzymes (DUBs) that counteract ubiquitination. USP15 and USP8 are the only DUBs identified to target p62. USP15 reverses the polyubiquitination of p62 by RNF26 in the endocytic pathway [26] whereas USP8 removes lysine 11-linked ubiquitin chains from p62 to suppress autophagic activity [27].

The homeotic balance between cell survival and death eventually determine the outcome of cell fate such as tumor formation and one of the critical regulators in this process is nuclear factor-κB (NF-κB). Two decades ago, the p62 protein was reported to bind to the RIPK1 protein, which connects the aPKCs to the TNFα receptor, resulting in NF-κB activation by the TNFα signaling pathway [12]. In addition to the TNFα signal, p62 was shown to be an important protein in NF-κB activation by IL-1 through interacting with TRAF6 [13]. Although these data emphasize the importance of p62 in NF-κB-mediated cell survival, the DUBs targeting p62 in the regulation of TNFα-induced NF-κB signaling pathway are yet unknown.

In this study, we demonstrate that ubiquitin-specific protease 20 (USP20) regulates the p62 protein in the TNFα-induced NF-κB signaling pathway. This is the first report that reveals the molecular mechanism of the USP20-p62 axis regarding NF-κB-mediated cell survival.

## 2. Results

### 2.1. USP20 Stabilizes the p62 Protein

Previously, we found that expression of the p62 autophagic adaptor protein shows slightly decreased basal levels without chloroquine in ubiquitin-specific protease 20 (USP20)-depleted HeLa cells, suggesting that USP20 directly regulates p62 expression [28]. To verify this possibility, we examined the expression of p62 mRNA or protein in USP20-depleted HeLa cells, compared to control cells expressing scrambled siRNAs (siCON). USP20 depletion did not affect the levels of p62 mRNA (Figure 1A), but significantly decreased the expression of p62 protein (Figure 1B). Immunofluorescence assays also showed that USP20 depletion in HeLa cells significantly reduces the expression of endogenous p62 protein (Figure 1C). Furthermore, dose-dependent augmentation of USP20 in HEK293 cells profoundly increased p62 expression (Figure 1D). These results suggest that the deubiquitinating enzyme USP20 is involved in the regulation of p62 protein stability and not p62 mRNA levels.

To further confirm these results, we examined the half-life of p62 protein in the presence of protein synthesis inhibitor cycloheximide (CHX). USP20 overexpression in HeLa cells significantly increased the half-life of p62 protein in the presence of CHX, together with the basal level of p62, but overexpression of a catalytically inactive (CI) mutant of USP20 did not (Figure 1E) [28]. In addition, the half-life of the p62 protein was markedly reduced in USP20-depleted HeLa cells compared to control HeLa cells, and the stability of basal level p62 was also affected (Figure 1F). Therefore, these results indicate that USP20 stabilizes the p62 protein.

### 2.2. USP20 Deubiquitinates p62K48-Linked Polyubiquitination

To understand the importance of USP20 deubiquitinase activity in regulating p62 stability, we first examined whether USP20 directly deubiquitinates p62. After plasmids encoding Flag-p62 and HA-ubiquitin (HA-Ubi) were co-transfected into HeLa cells, together with wild-type (WT) USP20 or catalytically inactive mutant USP20 (USP20-CI), immunoprecipitation and immunoblot assays were performed. Wild-type USP20 profoundly decreased the polyubiquitination of p62 whereas the catalytic inactive mutant did not (Figure 2A). Ni-NTA pull-down assays in HeLa cells also showed similar results, supporting that USP20 removes p62 polyubiquitination (Figure 2B).

Next, we investigated which polyubiquitination pattern of p62 is deubiquitinated by USP20. Wild-type ubiquitin (HA-Ubi), a K48 ubiquitin mutant (HA-Ubi-K48) in which six lysine residues except for lysine 48 are substituted into arginines, or a K63 ubiquitin mutant (HA-Ubi-K63) in which only lysine 63 is left intact, were transfected into HeLa cells with or without wild-type USP20 in the presence of Flag-p62. When the plasmid encoding the HA-Ubi-K48 mutant was transfected, cells were pre-treated with MG132 for 8 h to prevent the proteasomal degradation of Flag-p62. K48-linked polyubiquitination of p62 was significantly deubiquitinated by USP20, but K63-linked polyubiquitination was not affected (Figure 2C). Considering these results, it is evident that USP20 regulates p62 stability through deubiquitinating the K48-linked polyubiquitination chain.

### 2.3. USP20 is Required for TNFα-Induced NF-kB Activation through Direct Binding to p62

Our finding that USP20 stabilizes the p62 protein prompted us to examine whether USP20 directly binds to p62. Co-immunoprecipitation assays revealed the direct interaction of USP20 with p62 when plasmids encoding HA-USP20 and Flag-p62 were ectopically expressed in HeLa cells (Figure 3A). Based on this finding, we next examined the endogenous interaction of USP20 with p62 upon TNFα treatment. TNFα was chosen because the DUBs targeting p62 in TNFα-induced NF-κB activation are unknown, although p62 has been known to be involved in TNFα-induced NF-κB activation in HeLa cells through binding to aPKCs [12]. Immunoprecipitation assays supported the endogenous interactions between USP20 and p62 upon TNFα treatment in HeLa cells (Figure 3B). USP20 was observed to bind to p62 protein at 5 min post TNFα treatment and this complex subsequently dissociated (Figure 3B).

These results led us to investigate whether USP20 is required for TNFα-induced NF-κB activation. Ectopic expression of USP20 augmented NF-κB-mediated reporter activity upon TNFα treatment, and increased expression of p62 (Figure 3C). In addition, quantitative real-time RT-PCR analysis revealed that USP20 depletion decreases expression of NF-κB-mediated target genes such as *BFL1* and *cFLIP*, which are related to cell survival, upon TNFα treatment (Figure 3D). Furthermore, USP20 depletion significantly reduced IKKα/β phosphorylation and delayed IκBα degradation in the presence of TNFα (Figure 3E), which was similarly observed in p62-depleted HeLa cells (Appendix A).

Because our findings indicate that USP20 is required for TNFα-mediated NF-κB activation through targeting p62, we next examined whether the deubiquitination of endogenous p62 protein is regulated by USP20. In control HeLa cells expressing scrambled siRNAs (siCON), polyubiquitination of endogenous p62 was reduced at 5 min post TNFα treatment, with increased expression of p62 in total cell lysates, and this reduced polyubiquitination was subsequently recovered after 5 min (Figure 3F). This time point coincided with the timing of endogenous p62 interaction with USP20 (Figure 3B). In comparison to control cells, reduction of polyubiquitinated p62 was not observed at 5 min in USP20-depleted HeLa cells upon TNFα treatment, but the polyubiquitination pattern of endogenous p62 was increased (Figure 3F). These results provide robust evidence that USP20 is a positive regulator for the stabilization of p62 in TNFα-mediated NF-κB activation.

Considering these findings, it is possible that USP20 is a component of the signaling complex including p62, PKCζ and RIPK1 in TNFα-induced NF-κB activation, through its interaction with p62. To validate this possibility, we examined whether USP20 depletion disrupts formation of the signaling complex composed of p62, PKCζ and RIPK1 upon TNFα treatment. Immunoprecipitation assays against endogenous PKCζ in siCON-expressing control HeLa cells showed formation of the PKCζ-mediated signaling complex, whereas this signaling complex was disrupted in USP20-depleted cells (Figure 3G), indicating that USP20 is a crucial component in TNFα-induced NF-κB activation through deubiquitination and stabilization of the p62 protein.

### 2.4. USP20 Depletion Increases Apoptosis

The homeotic balance between cell survival and death determines cell fate and thus the reduction of cell survival is accompanied by increased cell death. Because our results suggest that the USP20-p62 axis is responsible for PKCζ-mediated NF-κB activation for cell survival, USP20 depletion may promote cell death. To verify this possibility, we examined cell viability in USP20-depleted and control HeLa cells in the presence of TNFα plus cycloheximide (CHX). The reason for using TNFα/CHX is because the response of mammalian cells to TNFα can be switched to apoptosis by co-treatment with CHX [29] and this apoptotic effect is independent on the RIPK1 protein [30]. USP20 depletion in HeLa cells significantly decreased cell viability and cell numbers in the presence of TNFα/CHX, compared to siCON-expressing control cells (Figure 4A–C). This apoptotic effect upon USP20 depletion was also observed in p62-depleted HeLa cells under the same conditions (Figure 4A–C). In addition, immunoblot analysis showed that USP20 depletion increases the expression of cleaved caspase-8, caspase-3 and PARP in the presence of TNFα/CHX, compared to control HeLa cells, as well as causing increased basal levels of these cleaved proteins without TNFα/CHX (Figure 4D). FACS analysis was consistent with the results that UPS20 depletion, causing reduction of the NF-κB-mediated pro-survival signal, increases RIPK1-independent apoptosis (Figure 4E).

Next, we examined whether USP20 depletion is involved in RIPK1-dependent apoptosis. To this end, we first examined cell viability and apoptosis of USP20-depleted and control HeLa cells upon treatment of TNFα plus a TAK1 inhibitor (5Z-7; 5z-7-oxozeaenol). When phosphorylation of RIPK1 is blocked by TAK1 deficiency or the TAK1 inhibitor, TNFα promotes the activation of RIPK1, resulting in RIPK1-dependent apoptosis [30]. Similar to TNFα/CHX treatment, USP20 depletion in HeLa cells as well as p62 depletion reduced cell viability and cell numbers in the presence of TNFα and 5Z-7 (Figure 5A–C). Immunoblot analysis clearly showed increased expression of cleaved caspase-8, caspase-3 and PARP in USP20-depleted HeLa cells upon treatment with TNFα/5Z-7 (Figure 5D). FACS analysis also revealed augmented apoptosis in USP20-depleted cells upon treatment with TNFα/5Z-7, compared to control HeLa cells. Thus, these findings clearly indicate that USP depletion, blocking NF-κB-mediated cell survival, increases cell death programs including both RIPK1-dependent and independent pathways.

However, we did not exclude the possibility that these apoptotic effects may be due to defects in autophagy by USP20 depletion upon TNFα/CHX or TNFα/5Z-7 treatment, because USP20 is involved in autophagic initiation [28]. To test this possibility, USP20-depleted and control HeLa cells were pre-treated with the autophagy inhibitor bafilomycin A1 (BafA1) and we subsequently examined apoptosis upon TNFα/CHX or TNFα/5Z-7 treatment compared with cells in the absence of BafA1. The blockade of autophagy by BafA1 did not affect the apoptosis induced by TNFα/CHX or TNFα/5Z-7 treatment (Appendix A), suggesting that autophagy is not involved in the apoptosis caused by USP20 depletion upon TNFα/CHX or TNFα/5Z-7 treatment.

These results promptly led us to examine whether USP20 depletion is related to necroptosis, which is a unique cell death program induced by TNFα under certain cell contexts that acts in a caspase-independent manner. To verify this possibility, we generated a USP20-depleted HT29 colon cancer cell line by infection of lentiviruses expressing USP20-specific shRNA. HT29 colon cancer cells are widely used to detect necroptosis by co-treatment with TNFα, BV6 (IAP inhibitor) and z-VAD-fmk (caspase inhibitor) [31], and MLKL (mixed lineage kinase domain like pseudokinase) phosphorylation is used as a marker of necroptosis [32]. USP20 depletion in HT29 cells did not affect MLKL phosphorylation in the presence of TNFα/BV6/z-VAD-fmk, compared to control cells (Appendix A). In addition, UPS20 depletion did not cause any remarkable changes in cell morphology and cell death (Appendix A), suggesting that USP20 depletion is not required for necroptosis in this experimental condition.

## 3. Discussion

Ever since USP20 was identified as a substrate of the cullin-RING ligase family member CRL^VHL^ [33,34], accumulating evidence has revealed that USP20 is involved in diverse cellular signaling pathways such as hypoxia, the DNA damage response, TLR4-mediated innate immune response, the mitogen-signaling pathway and autophagy [13,28,35,36,37,38]. In this study, we demonstrate that USP20 is a positive regulator in TNFα-induced cell survival through stabilizing the p62 protein. Although the p62 protein has been recognized as a signaling hub in PKCζ-mediated NF-κB activation upon TNFα treatment [12], about the mechanisms of the ubiquitin modifications of the p62 protein in this pathway have remained unknown. This is the first report to identify a deubiquitinating enzyme involved in post-translational modification of p62 regarding PKCζ-mediated NF-κB activation in the presence of TNFα and demonstrate the molecular mechanism that regulates stability of the p62 protein

While much attention about the p62 protein has been given to autophagy, our previous study suggested the possibility that USP20 regulates p62 involved in other pathways as well as autophagy [28]. Based on this speculation, our current experiments provide evidence that the p62-USP20 axis is required for PKCζ-mediated NF-κB activation upon TNFα treatment. Atypical PKCζ has been known to be connected to RIPK1 through the p62 protein in NF-κB activation, resulting in formation of the PKCζ-p62-RIPK1 signaling complex [12]. However, the deubiquitinating enzymes involved in this pathway have not been identified. Herein, we find that USP20 binds to p62 at 5 min upon TNFα treatment and thus stabilizes it through removing the lysine 48 (K48)-linked polyubiquitin chains. This kinetic was consistent with IκBα degradation observed upon TNFα treatment. In addition, USP20 depletion significantly decreased IKKα/β phosphorylation and delayed IκBα degradation, resulting in the reduction of NF-κB-mediated cell survival. These results were caused by impairment of the scaffolding activity of p62, due to p62 instability caused by USP20 depletion. Disruption of the PKCζ-p62-RIPK1 complex upon TNFα treatment in USP20-depleted cells strongly supported the impairment of p62 scaffolding activity. Therefore, our findings re-emphasize the importance of the p62 protein as a scaffold molecule in PKCζ-mediated NF-κB activation and reveal a new post-translational mechanism for regulating p62 scaffolding activity.

Based on our data that USP20 is responsible for NF-κB-mediated cell survival through stabilizing p62, we also suggest that USP20 depletion causes the blockade of TNFα-induced NF-κB activation regarding PKCζ and thus may accelerate cell death pathways. Our present findings reveal that USP depletion increases both RIPK1-independent and RIPK1-dependent apoptosis, which has previously been suggested [30], depending on stimuli such as TNFα/CHX and TNFα/5Z-7, respectively. Accumulating evidence has shown that the underlying molecular mechanisms of the TNF signaling pathway are very complicated with regards to cell survival, apoptosis and necroptosis [32]. Binding of TNFα to TNF receptor 1 (TNFR1) causes interaction with TNFR1-associated death domain protein (TRADD), which subsequently recruits RIPK1, TNFR-associated factors (TRAF) and cellular inhibitor of apoptosis proteins (cIAPs), resulting in the formation of Complex І, which is required for cell survival. The transition from cell survival to apoptosis in the presence of TNFα needs the release of non-ubiquitinated RIPK1 from Complex І. The release of RIPK1 facilitates the formation of a second protein complex (Complex ІІa or ІІb) composed of Fas-associated death domain protein (FADD), pro-caspase 8 and the long isoform of FLICE-like inhibitory protein (FLIPL). Thus, the release of deubiquitinated RIPK1 plays a crucial role in the decision of cell fate upon TNFα treatment. In addition, a recent study indicated the importance of RIPK1 phosphorylation status by TAK1 in the apoptotic pathway [30]. These findings are likely to be compatible with our results that USP20 depletion in HeLa cells increases apoptosis upon TNFα/CHX or TNFα/5Z-7 treatment.

We speculate that the USP20-p62 axis works on the upstream of RIPK1 in non-classical PKCζ-mediated NF-κB activation upon TNFα treatment and USPS20 depletion decreases p62 scaffolding activity, resulting in the release of RIPK1. In fact, our present results that USP20 depletion disrupts PKCζ-p62-RIPK1 may support this speculation. Therefore, in a USP20-depleted background, released RIPK1 may or may not be phosphorylated by TAK1 depending on cell context, eventually contributing to RIPK1-dependent or RIPK1-independent apoptosis. In contrast, it may be interesting that USP20 depletion does not affect the unique cell death program necroptosis. Although we do not understand the exact reasons, it is possible that non-classical PKCζ-mediated NF-κB activation upon TNFα treatment is not related to necroptosis under our current conditions.

Furthermore, our finding that USP20 removes the polyubiquitin chains of p62 suggests that a certain E3 ubiquitin ligase facilitates the K48-linked polyubiquitination of p62 in PKCζ-mediated NF-κB activation upon TNFα treatment. Thus, future identification of the specific E3 ubiquitin ligase that acts on the p62 protein should yield further insight into the molecular mechanisms of ubiquitination and deubiquitination of p62 in PKCζ-mediated NF-κB activation.

In conclusion, we here demonstrate that a deubiquitinating enzyme USP20 is required for the PKCζ-mediated NF-κB signaling pathway through stabilizing the p62 scaffolding protein in the presence of TNFα. Thus, our data define a new mechanism of post-translational modification of p62 by USP20 in the TNFα signaling pathway.

## 4. Materials and Methods

### 4.1. Cell Culture and Reagents

HEK293T (human embryonic kidney), HeLa (human cervix adenocarcinoma) and HT29 (human colorectal adenocarcinoma) cell lines were purchased from the American Type Culture Collection (ATCC, Manassas, VA, USA). HEK293T and HeLa cells were cultured in DMEM with 10% FBS (Hyclone, Logan, UT, USA). HT29 cells were maintained with McCoy’s 5a medium (Thermo Fisher Scientific, Waltham, MA, USA) with 10% FBS. Recombinant human TNFα (210-TA) was purchased from R&D systems (Minneapolis, MN, USA). Cycloheximide (C4859) was purchased from Sigma-Aldrich (St. Louis, MO, USA). BV6 and z-VAD-fmk were obtained from Selleckchem (Houston, TX, USA). 5z-7-oxozeaenol used as a TAK1 inhibitor was purchased from Tocris (Bristol, UK).

### 4.2. RNA Extraction and Quantitative Real-Time RT-PCR

Isolation of total RNA of cell lines and reverse transcription were performed as previously described [39]. Primer sequences of each TNFα target gene and *Gapdh* are described in Appendix A. For quantitative real-time RT-PCR, we used an iCycler real-time PCR machine and iQ SYBR Green Supermix (Bio-Rad, Hercules, CA, USA) to measure gene expression levels. Expression of the genes was measured under the following conditions: 40 cycles of 95 °C for 30 s, 60 °C for 30 s and 72 °C for 30 s. All reactions were independently repeated at least three times for reproducibility of experiments.

### 4.3. Immunoblotting and Immunoprecipitation

For immunoblot assay, lysis buffer (1% Triton X-100, 50 mM Tris-HCl at pH 7.5, 150 mM NaCl, 10% glycerol, protein inhibitor cocktail, 10 mM NaF, 1 mM NaOV) was used to lyse cells. 20 μg of proteins from whole cell extract per sample were separated by sodium dodecyl sulfate-polyacrylamide gel electrophoresis (SDS-PAGE) with Tris-glycine buffer containing 0.1% SDS. Subsequently, the separated proteins were transferred to a 0.45-μm PVDF membrane filter (Merk Millipore, Burlington, MA, USA) and immunoblot analysis performed. For immunoprecipitation, protein lysates were incubated with the indicated antibodies and G agarose beads (GenDEPOT, Katy, TX, USA) at 4 °C for 15 h. Protein lysates were washed three times with lysis buffer and separated from the beads by boiling 5 min with 2x sample buffer. The following antibodies were used for immunoblotting and immunoprecipitation analysis. Anti-HA-HRP (catalog number, 12013819001; dilution ratio, 1:5000) and anti-Ubiquitin FK2-HRP (BMLPW0150; 1:5000) antibodies were purchased from Roche (Basel, Switzerland) and Enzo Life Sciences (Farmingdale, NY, USA), respectively. Anti-p-MLKL (ab187091; 1:2000) and anti-USP20 (A301-189A; 1:2000) antibodies were obtained from Abcam (Cambridge, UK) and Bethyl Lab (Montgomery, TX, USA), respectively. Rabbit anti-phospho-IкBα (2859; 1:2000), mouse anti-IкBα (4814; 1:2000), rabbit anti-phospho-IKKα/β (2697; 1:2000), mouse anti-IKKα (11930; 1:2000), rabbit anti-IKKβ (8943; 1:2000), mouse anti-p62 (88588; 1:4000), rabbit anti-PARP (9542; 1:5000), mouse anti-Caspase8 (9746; 1:2000), rabbit anti-Cleaved-Caspase3 (9664; 1:2000), rabbit anti-Caspase3 (9662; 1:2000), rabbit anti-TRADD (3694; 1:2000), rabbit anti-RIP1 (3493; 1:2000), rabbit anti-MLKL (14993; 1:2000) and rabbit anti-K48-linkage specific polyubiquitin (12805; 1:2000) antibodies were purchased from Cell Signaling Technology (Danvers, MA, USA). Mouse anti-TNFR1 (sc-8436; 1:2000) and mouse anti-PKCζ (sc-17781; 1:2000) antibodies were purchased from Santa Cruz Biotechnology (Dallas, TX, USA).

### 4.4. Immunofluorescence Assays

For immunofluorescence assay, cells were fixed by cold methanol at −20 °C for 7 min, followed by blocking (5% BSA in) at room temperature for 30 min and incubation with primary antibodies at 4 °C for 12 h. Mouse anti-p62 (Cell Signaling Technology; catalog number, 88588; dilution ratio, 1:400) primary antibody was used in this assay. After washing with PBS five times, coverslips were stained with the following secondary antibodies at room temperature for 2 h: Alexa Fluor-488-conjugated goat anti-mouse IgG (Invitrogen, Carlsbad, CA, USA, 1:800 for anti-p62). Coverslips were stained with DAPI (Sigma-Aldrich) and mounted on glass slides. Cells were examined with a laser scanning confocal microscope (Carl-Zeiss, Oberkochen, Germany).

### 4.5. Plasmids

Human p62 cDNA was amplified by PCR from the cDNAs of HeLa cells. Full length human p62 cDNA was cloned into the *EcoR*I and *Xho*I site of pcDNA3-HA and pcDNA3-Flag vector (Forward: 5′-CCGGAATTCATGGCGTCGCTCACCGTG-3′, Reverse: 5′-CCGCTC-GAGTCACAACGGCGGGGGATG-3′). Wild-type USP20 and its point mutant plasmids and 5xNF-kB-Luc luciferase reporter plasmid were previously described [28,40].

### 4.6. Transfection of Plasmid and siRNAs

Plasmids were transiently transfected into HEK293 cells using PEI (Polyethyleneimine) and HeLa cells using Lipofectamine 2000 (Invitrogen) according to the manufacturer’s instructions. Human p62 and USP20 siRNA oligonucleotides were synthesized by Genolution (Seoul, Korea) and their sequences are described in Appendix A. Each siRNA was reverse-transfected using Lipofectamine RNAi-MAX (Invitrogen) according to the manufacturer’s instructions.

### 4.7. In Vivo Ubiquitination Assay

To perform in vivo ubiquitination assays, lysis buffer Ⅰ [PBS containing Tris-HCl at pH 7.5, 1% SDS, 5 mM *N*-ethyl maleimide (NEM), protein inhibitor cocktail] was added to cells for harvest. Subsequently, samples were boiled for 10 min to dissociate non-covalent protein interaction. After 10 times dilution with lysis buffer Ⅱ (PBS containing Tris-HCl at pH 7.5, 1% TritonX-100, 5 mM NEM), samples were suspended by 1 mL syringe. The lysates were subsequently separated from debris by centrifugation at 13,000 rpm for 10 min. Lysates were incubated at 4 °C for 15 h with indicated antibodies and G agarose beads. The beads were washed three times with lysis buffer Ⅱ and samples were boiled for 5 min with 2x sample buffer. Immunoprecipitation samples were transferred onto PVDF membranes and the membranes were denatured by 6-M guanidine hydrochloride buffer (20-mM Tris-HCl pH7.5 buffer containing 6-M guanidine-HCl, 5-mM β-mercaptoethanol) at 4 °C for 30 min. After membranes were washed with washing buffer three times, they were blocked by 5% BSA for 2 h and incubated with anti-FK2-HRP antibody (Enzo Life Sciences, BML-PW9910) at 4 °C overnight to detect ubiquitination patterns.

### 4.8. Pull-Down and Ubiquitination Assay by Ni-NTA columns

Cells were collected in PBS buffer containing 5 mM NEM. Cells were resuspended in binding buffer (6-M guanidine-HCl, 0.1-M Na_2_HPO_4_, 0.1-M NaH_2_PO_4_, 0.01-M Tris-HCl (pH 8.0), 10 mM β-mercaptoethanol, 5 mM NEM, 5 mM imidazole) and incubated with Ni-NTA agarose (Qiagen, Hilden, Germany) at 4 °C for 12 h. Ni-NTA-mediated pull-down assays were performed as described [41].

### 4.9. Construction of Small Hairpin RNAs and Lentiviral Infection

The specific small hairpin RNAs (shRNAs) were obtained from Mission-shRNA (Sigma-Aldrich). The sequences for USP20-specific shRNA were as followed; shUSP20 #3: 5′- CCGGCTATGTTGGCTGCGGAGAATCCTCGAGGATTCTCCGCAGCCAACATAGTTTTTG -3′. A lentiviral packaging system purchased from Invitrogen was used to generate a lentivirus expressing each shRNA. Green fluorescence protein (GFP)-targeting shRNA was used as a negative control for lentiviral infection. Generation of recombinant lentiviruses was performed according to the protocols previously described [42].

### 4.10. Apoptosis and MTT Analysis

Cells were gathered using trypsin and then resuspended in Annexin V binding buffer. 1 × 10^5^ cells in a 100 μL volume were incubated with Annexin V-APC and propidium iodide (BD Biosciences, San Jose, CA, USA) for 15 min at room temperature in the dark. 100 μL of 1x binding buffer was added and fluorescence was detected using FACS Canto II (BD Biosciences). To perform the MTT assay, USP20-depleted, p62-depleted and control HeLa cells were cultured in six-well plates. MTT solution (114650-070-01; Sigma-Aldrich) was added to each well and incubated for 20 m at 37 °C according to the manufacturer’s instruction. The media was discarded and 2 mL of DMSO was subsequently added. Absorbance values at 490 nm were determined by a Bio-Rad 680 microplate reader (Bio-Rad).

### 4.11. Statistical Analysis

Quantitation of immunoblotted proteins was performed using ImageJ software [43]. Data are expressed as mean ± SD. Statistical significance was calculated by one-way or two-way ANOVA using GraphPad Prism 8 software. *p* < 0.05 was considered statistically significant. All results shown in figures are representative of at least three independent experiments.

## Figures and Tables

**Figure 1 ijms-21-03116-f001:**
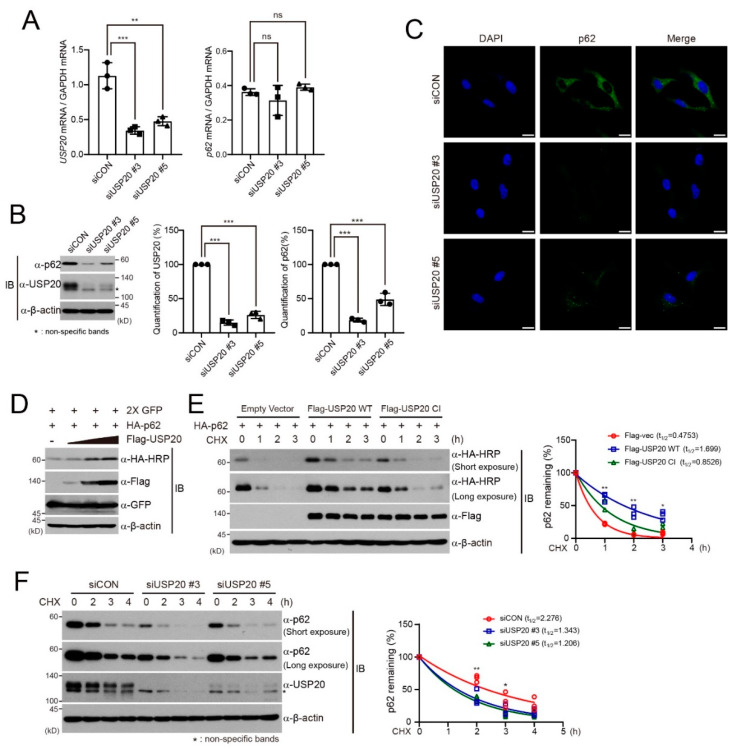
USP20 is required for p62 protein stability. (**A**) HeLa cells were reverse-transfected with 20-nM control siRNA (siCON) or one of two independent USP20-specific siRNAs (siUSP20 #3 and siUSP20#5). *USP20* and *p6*2 mRNA levels were measured by quantitative real-time reverse -transcriptase-polymerase chain reaction (qRT-PCR) in control and USP20-depleted HeLa cells and normalized to *Gapdh* mRNA. (**B**) Total cell lysates obtained from USP20-depleted and control HeLa cells were immunoblotted with the indicated antibodies. Expression levels of endogenous p62 and USP20 proteins were quantified by using ImageJ software. For normalization, β-actin expression was used as a control. (**C**) Endogenous expression of p62 protein in USP20-depleted and control HeLa cells were detected by immunofluorescence analysis. Scale bars, 20 µm. (**D**) After a plasmid encoding HA-p62 was co-transfected into HEK293 cells with dose-dependent expression of Flag-USP20, cells were immunoblotted with the indicated antibodies. Expressions of green fluorescence protein (GFP) and β-actin were used as loading controls. (**E**) After plasmids encoding wild-type (WT) Flag-USP20 or a catalytically inactive (CI) mutant of Flag-USP20 were respectively transfected into HeLa cells, cells were treated with 50 μg/mL cycloheximide (CHX) for the indicated times and lysates were immunoblotted with the indicated antibodies. Empty vector was used as a control. (**F**) USP20-depleted or control (siCON) HeLa cells were treated with 50 μg/mL CHX for the indicated times. Total cell lysates were immunoblotted with the indicated antibodies. In (**E**) and (**F**), p62 levels were quantified by ImageJ software and normalized to β-actin expression. Data were statistically analyzed by two-way ANOVA followed by Bonferroni’s multiple comparison test (* *p* < 0.05, ** *p* < 0.01, *** *p* < 0.001 compared to siCon or empty vector, ns; not significant, *n* = 3). Bars represent the mean ± SD. Images are representative of three independent experiments.

**Figure 2 ijms-21-03116-f002:**
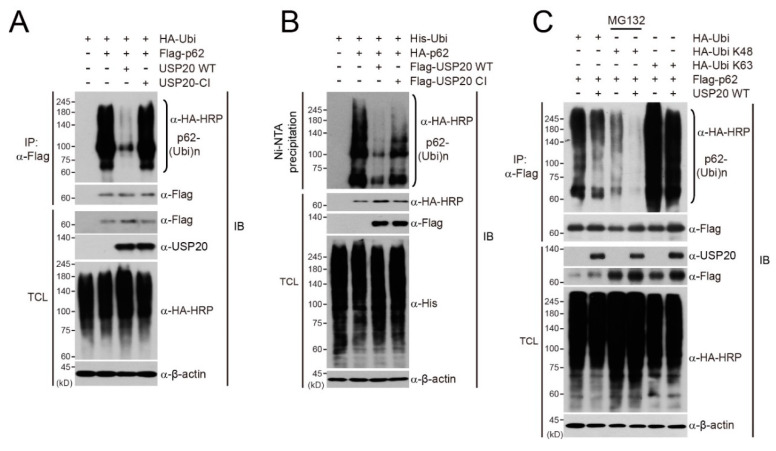
USP20 removes the polyubiquitin chains of p62. (**A**) A catalytically inactive (CI) mutant of USP20 (USP20-CI) or wild-type USP20 (USP20-WT) were co-transfected into HeLa cells with Flag-p62 and HA-Ubi in the indicated combinations. p62 ubiquitination was examined by immunoprecipitation (IP) and immunoblots (IB) with the indicated antibodies. (**B**) A catalytically inactive (CI) mutant of Flag-USP20 (Flag-USP20-CI) and wild-type Flag-USP20 (Flag-USP20-WT) were co-transfected into HeLa cells with Flag-p62 and His-Ubi in the indicated combinations. Ni-NTA-mediated pull-down assays were performed and ubiquitinated p62 was observed by immunoblotting using anti-HA antibody. (**C**) Flag-p62 was co-transfected into HeLa cells with wild-type HA-Ubi or a lysine mutant (K48 or K63) of HA-Ubi in the absence or presence of wild-type USP20. p62 ubiquitination was examined by immunoprecipitation (IP) and immunoblots (IB) with the indicated antibodies. When the plasmid encoding the K48 lysine mutant of HA-Ubi was transfected, cells were pre-treated with MG132 for 8 h to prevent p62 degradation. Expression of β-actin was used as a loading control. The images in immunoblot analysis are representative of three independent experiments. TCL; total cell lysates.

**Figure 3 ijms-21-03116-f003:**
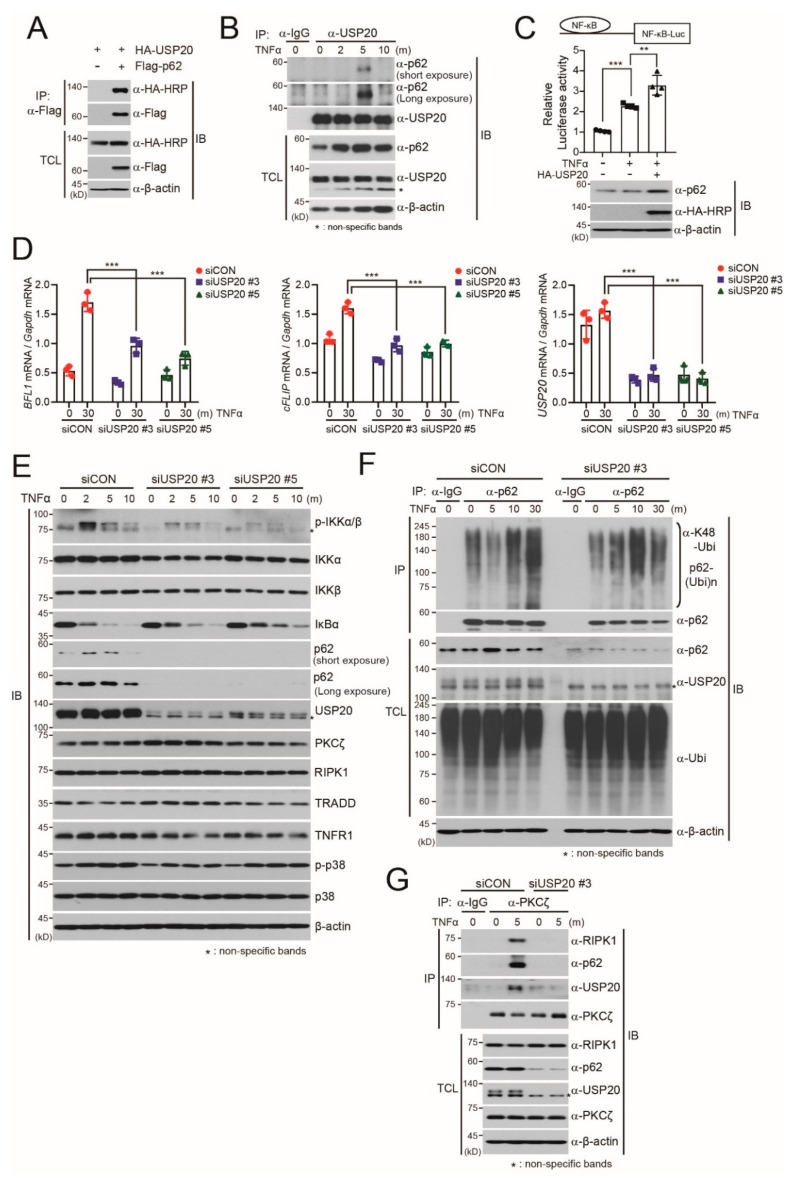
USP20 is crucial for PKCζ-mediated NF-κB activation upon TNFα treatment through direct binding to p62. (**A**) Plasmids encoding HA-USP20 were co-transfected into HeLa cells with Flag-p62 plasmid. Cell lysates were immunoprecipitated (IP) with anti-Flag antibody and subsequently immunoblotted (IB) with anti-HA or anti-Flag antibodies. TCL; total cell lysates. (**B**) After HeLa cells were treated with 10 ng/mL TNFα, endogenous p62 was immunoprecipitated with anti-p62 antibody and subsequently immunoblotted with the indicated antibodies against endogenous USP20 and p62. Anti-IgG antibody was used a negative control for IP. Expression of β-actin was used as a loading control. (**C**) After a plasmid encoding a NF-κB-mediated luciferase reporter (NF-κB-Luc) was co-transfected into HeLa cells in the absence or presence of HA-USP20, cells were treated with 10 ng/mL TNFα for 90 min and luciferase assays were subsequently performed. Luciferase activity was normalized to the expression of *Renilla* luciferase. Expression of USP20 and endogenous p62 protein was confirmed by immunoblots. (**D**) Total RNA was respectively isolated from two independent USP20-depleted and control HeLa cells at the indicated time points after 10 ng/ ml TNFα treatment. Expression of *BFL1*, *cFLIP* and *USP20* mRNA was monitored by quantitative RT-PCR analysis and normalized to the expression of *Gapdh* mRNA. The data in (**C**) and (**D**) were statistically analyzed by two-way ANOVA followed by Bonferroni’s multiple comparison test (^**^
*p* < 0.01, ^***^
*p* < 0.001 compared to the indicated controls, *n* = 3). The bars indicate the mean ± SD. (**E**) HeLa cells were reverse-transfected with 20-nM control siRNA (siCON) or one of two independent USP20-specific siRNAs (siUSP20 #3 and siUSP20#5) and treated with TNFα for the indicated times. Total cell lysates were immunoblotted with the indicated antibodies. (**F**) For the ubiquitination assay of endogenous p62 protein, USP20-depleted and control HeLa cells were treated with 10 ng/mL TNFα for the indicated times. Cell lysates were immunoprecipitated with anti-p62 antibody under 1% SDS denaturing conditions and subsequently immunoblotted with anti-ubiquitin antibody. Total cell lysates were immunoblotted by the indicated antibodies. As a negative control for IP, cell lysates were immunoprecipitated with anti-IgG antibody. Expression of β-actin and total ubiquitin was used as loading controls. **G:** To examine whether USP20 depletion influences the formation of the TNFα-induced RIP1-p62-PKCζ signaling complex, USP20-depleted and control HeLa cells were treated with 10 ng/mL TNFα for the indicated times. Cell lysates were immunoprecipitated with anti-PKCζ antibody and subsequently immunoblotted with the indicated antibodies. Anti-IgG antibody was used as a negative control for IP. Expression of β-actin was used as a loading control. The immunoblot images in this figure are representative of three independent experiments.

**Figure 4 ijms-21-03116-f004:**
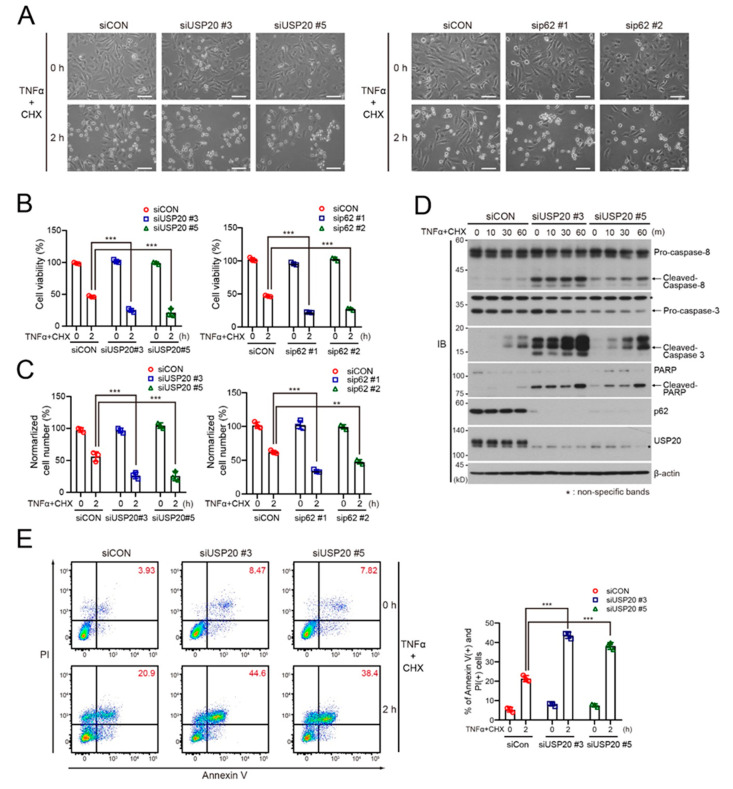
USP20 depletion is involved in TNFα-induced RIPK1-independent apoptosis. (**A**–**C**) USP20-depleted, p62-depleted and control HeLa (siCON) cells were treated with 20 ng/mL TNFα plus 10 μg/mL cycloheximide (CHX) for 2 h. Morphologic changes were observed by light microscopy (**A**). Scale bars, 1000 μm. MTT assay (**B**) and live cell counting (**C**) were performed to measure cell viability. Data were statistically analyzed by two-way ANOVA followed by Bonferroni’s multiple comparison test (** *p* < 0.01, *** *p* < 0.001 compared to control cells without TNFα/CHX, *n* = 3). The bars represent the mean ± SD. (**D**) USP20-depleted and control HeLa cells were treated with 20 ng/mL TNFα plus 10 μg/mL cycloheximide (CHX) for the indicated times. Cell lysates were immunoblotted with the indicated antibodies. (**E**) USP20-depleted and control HeLa cells were treated with 20 ng/mL TNFα plus 10 μg/mL cycloheximide (CHX) for 2 h. Cells were immediately stained with Annexin V and propidium iodide (PI), followed by FACS analysis. The percentage of Annexin V-positive and PI-positive cells was summarized in a bar graph. The data were statistically analyzed by two-way ANOVA followed by Bonferroni’s multiple comparison test (^***^
*p* < 0.001 compared to the control cells without TNFα/CHX, *n* = 3). Bars indicate the mean ± SD. The images in (**A**) and (**D**) are representative of three independent experiments.

**Figure 5 ijms-21-03116-f005:**
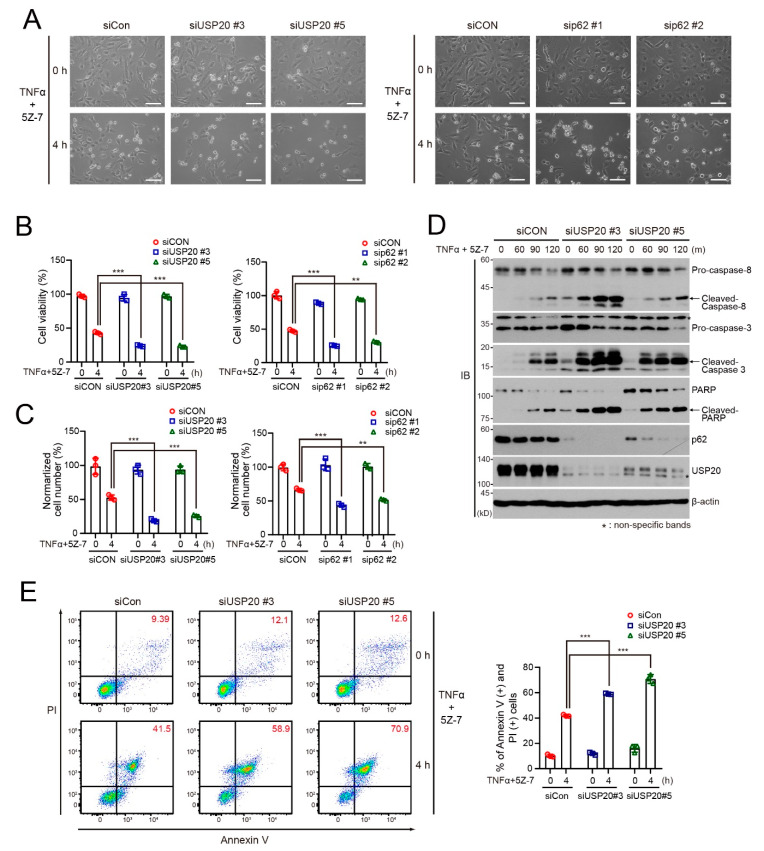
USP20 depletion is involved in TNFα-induced RIPK1-dependent apoptosis. **A**–**C**: USP20-depleted, p62-depleted and control HeLa (siCON) cells were treated with 20 ng/mL TNFα plus 1-μm 5Z-7 (TAK1 inhibitor) for 4 h. Morphologic changes were observed by light microscopy (**A**). Scale bars, 1000 μm. MTT assay (**B**) and live cell counting (**C**) were performed to measure cell viability. Data were statistically analyzed by two-way ANOVA followed by Bonferroni’s multiple comparison test (** *p* < 0.01, *** *p* < 0.001 compared to the control cells without TNFα/5Z-7, *n* = 3). Bars represent the mean ± SD. (**D**) USP20-depleted and control HeLa cells were treated with 20 ng/mL TNFα plus 1-μm 5Z-7 for the indicated times. Cell lysates were immunoblotted with the indicated antibodies. (**E**) USP20-depleted and control HeLa cells were treated with 20-ng/mL TNFα plus 1-μm 5Z-7 for 4 h. Cells were immediately stained with Annexin V and propidium iodide (PI), followed by FACS analysis. The percentage of Annexin V-positive and PI-positive cells was summarized in a bar graph. The data were statistically analyzed by two-way ANOVA followed by Bonferroni’s multiple comparison test (*** *p* < 0.001 compared to control cells without TNFα/5Z-7, *n* = 3). Bars represent the mean ± SD. The images in (**A**) and (**D**) are representative of three independent experiments.

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
