# Peer review of "The Deubiquitinating Enzyme USP20 Regulates the TNFα-Induced NF-κB Signaling Pathway through Stabilization of p62"

_ijms, 2020, doi:10.3390/ijms21093116_

Round 1
Reviewer 1 Report
The authors demonstrated a new DUB, ubiquitin-specific protease 20 (USP20), regulating the p62 protein in TNFa-induced NF-kB signaling pathway. This is the first report that reveals the molecular mechanism of USP20-p62 axis regarding NF-kB-mediated cell survival.
The data are potentially interesting and worthy of eventual publication.
This study is interesting, but several points should be clarified are needed prior to publication.
1)Please describe the protein concentration used in immunoblotting and immunoprecipitation.
2) Need more description / SDS-PAGE
3)Line 37 “ In particular, p62 is primarily degraded through selective autophagy [5, 6], although it can be degraded by the proteasome or endosomal-related autophagy [7-9], and thus its level is commonly measured to monitor autophagic flux. “ should be cited with a few more reference.
Author Response
Q1)Please describe the protein concentration used in immunoblotting and immunoprecipitation.
Answer: In this study, we used 20 μg of proteins from whole cell extract per sample in all immunoblot experiments to detect the expressions of specific proteins in total cell lysates. In the revised manuscript, the protein concentration used in immunoblot analysis was described in the “Materials and Methods” section (line 386, p12).
The protein concentrations immunoprecipitated by antibodies were difficult to measure because the amounts of immunoprecipitated proteins were variable depending on the quality of each antibody. Therefore, we used the same amounts of total cell lysate per sample in the initiation of immunoprecipitation experiments and initial protein concentrations were experimentally determined depending on the antibodies. Because of these reasons, we could not describe the protein concentrations used for immunoprecipitation analysis.
Q2) Need more description / SDS-PAGE
Answer: In the revised manuscript, the following sentences regarding SDS-PAGE were described in the “Materials and Methods” section (line 386-388, p12); 20 μg of proteins from whole cell extract per sample were separated by sodium dodecyl sulfate-polyacrylamide gel electrophoresis (SDS-PAGE) with Tris-glycine buffer containing 0.1 % SDS.
Q3)Line 37 “ In particular, p62 is primarily degraded through selective autophagy [5, 6], although it can be degraded by the proteasome or endosomal-related autophagy [7-9], and thus its level is commonly measured to monitor autophagic flux. “ should be cited with a few more reference.
Answer: We added two new references regarding the sentence pointed out in the revised manuscript (line 40, p1) and also edited the reference numbers, indicated in red, in the revised manuscript.
Reviewer 2 Report
Ha et al. in this manuscript identified USP20 as a deubiquitinating enzyme that regulates p62 polyubiquitination and stability. Moreover they show that USP20-p62 signalling is required for NF-kB activation following TNFa treatment, supporting cell survival.
This is a well conducted and elegant study. All methods used are properly described. Results are clearly presented and detailed, and more importantly the conclusions are strongly supported by the results.
However, The language is very poor, grammatically incorrect with particular reference to Abstract, Introduction and Conclusion. The manuscript needs a strong language revision.
Overall I suggest its publication but only an extensive editing.
Author Response
Q1) Overall I suggest its publication but only an extensive editing.
Answer: We edited the grammar and sentences of our manuscript using “Track change” function with the help of a native English-speaking scientist.
Round 2
Reviewer 1 Report
Accept in present form